# On the Robustness of the MnSi Magnetic Structure Determined by Muon Spin Rotation

**Pierre Dalmas de Réotier** [1,*,†] **, Alain Yaouanc** [1,†] **, Alex Amato** [2] **, Alexander Maisuradze** [3], **Daniel Andreica** [4] **, Bertrand Roessli** [5] **, Tatsuo Goko** [2] **, Robert Scheuermann** [2] **and Gérard Lapertot** [1]

1. Institut Nanosciences et Cyrogénie, University Grenoble Alpes, CEA, Pheliqs, 38000 Grenoble, France; alain.yaouanc@cea.fr (A.Y.); gerard.lapertot@cea.fr (G.L.)
2. Laboratory for Muon-Spin Spectroscopy, Paul Scherrer Institute, 5232 Villigen-PSI, Switzerland; alex.amato@psi.ch (A.A.); tatsuo.goko@gmail.com (T.G.); robert.scheuermann@psi.ch (R.S.)
3. Department of Physics, Tbilisi State University, Chavchavadze 3, 0128 Tbilisi, Georgia; alexander.maisuradze.m@gmail.com
4. Faculty of Physics, Babes-Bolyai University, 400084 Cluj-Napoca, Romania; daniel.andreica@phys.ubbcluj.ro
5. Laboratory for Neutron Scattering and Imaging, Paul Scherrer Institute, 5232 Villigen-PSI, Switzerland; bertrand.roessli@psi.ch
* Correspondence: pierre.dalmas-de-reotier@cea.fr
† These authors contributed equally to this work.

**Abstract:** Muon spin rotation ($\mu$SR) spectra recorded for manganese silicide MnSi and interpreted in terms of a quantitative analysis constrained by symmetry arguments were recently published. The magnetic structures of MnSi in zero-field at low temperature and in the conical phase near the magnetic phase transition were shown to substantially deviate from the expected helical and conical structures. Here, we present material backing the previous results obtained in zero-field. First, from simulations of the field distributions experienced by the muons as a function of relevant parameters, we confirm the uniqueness of the initial interpretation and illustrate the remarkable complementarity of neutron scattering and $\mu$SR for the MnSi magnetic structure determination. Second, we present the result of a $\mu$SR experiment performed on MnSi crystallites grown in a Zn-flux and compare it with the previous data recorded with a crystal obtained from Czochralski pulling. We find the magnetic structure for the two types of crystals to be identical within experimental uncertainties. We finally address the question of a possible muon-induced effect by presenting transverse field $\mu$SR spectra recorded in a wide range of temperature and field intensity. The field distribution parameters perfectly scale with the macroscopic magnetization, ruling out a muon-induced effect.

**Keywords:** manganese silicide; chiral magnetism; helimagnet; conical magnetic phase; muon spin rotation

## 1. Introduction

The physics of the intermetallic compound MnSi has attracted much attention since its crystal structure was established at room temperature in 1933 [1]. It crystallizes in the cubic $P2_13$ space group characterized by the absence of a center of symmetry. This leads to the possible existence of a Dzyaloshinskii–Moriya interaction. This means that the magnetic ordering occurring below approximately $T_c \simeq 29$ K [2] can be chiral. This is effectively the case as found by neutron diffraction [3,4]. In zero field, the Mn magnetic moments form a left-handed helix with an incommensurate propagation vector **k** parallel to one of the four three-fold axes. When an external field **B**$_{\text{ext}}$ of sufficient strength is

applied, **k** aligns along $\mathbf{B}_{ext}$ and a moment component parallel to this field is superimposed onto the helical component. This is the conical phase. The interest in MnSi has been renewed in 2001 with the discovery of the non-Fermi-liquid nature of the paramagnetic phase when the transition temperature is tuned towards absolute zero by application of hydrostatic pressure [5]. Confirming the exotic nature of this phase, quasi-static magnetic moments survive far above $T_c$ [6]. Last but not the least, a magnetic skyrmion lattice has been unravelled in the so-called A phase of the temperature-magnetic field phase diagram [7].

Due to the small modulus of the **k** vector, usual neutron diffraction techniques are not suitable for determining the arrangement of the magnetic moments in the magnetic structure. In counterpart, the small angle neutron scattering (SANS) technique was instrumental for establishing the triangular nature of the skyrmion lattice and the crystal direction and modulus of **k** in zero field, for instance. However, the extracted information is limited to these results. This has motivated us to attempt the determination of the magnetic structure in zero magnetic field, as well as near $T_c$ in the conical phase, using the muon spin rotation ($\mu$SR) technique. Thanks to the previously established muon position in the MnSi crystal structure [8], and the available information on **k**, the magnetic structures have been resolved. We found unconventional magnetic order in the helical [9] and conical [10] phases.

In this paper, we focus our interest on the magnetic structure in the helical phase. Before recalling the deviation that was unravelled with respect to the conventional helical state, essential features of MnSi and some results of representation analysis [11] must be exposed. In MnSi, the Mn atoms occupy the 4*a* position in Wyckoff notation [12], meaning that there are four Mn sites in the cubic unit cell. The local symmetry at a given 4*a* site is one of the four three-fold $\langle 111 \rangle$ axes of the crystallographic structure. In zero-field, the magnetic propagation vector **k**, with $k = 0.35$ nm$^{-1}$, is parallel to a $\langle 111 \rangle$ axis. When **k** $\parallel \langle 111 \rangle$, representation analysis enforces that the four Mn sites split into two families called orbits. The Mn site whose local three-fold axis is parallel to **k** belongs to orbit 1, while orbit 2 contains the other three Mn sites for which the local symmetry axis is not parallel to **k**. Whereas the relative phase of the magnetic moments of atoms belonging to a given orbit is given by the scalar product **k** · **r**, where **r** denotes the atom location in the crystal, representation analysis imposes no phase relation for moments belonging to different orbits.

In the course of the considered $\mu$SR experiment, some $10^8$ muons are implanted in the sample to be studied, probing the magnetic field at their stopping site, an interstitial position in the crystallographic structure. In zero external field, the local magnetic field is the vectorial sum of the dipole field resulting from the dipolar interaction between the muon and Mn magnetic moments and the contact field associated with the polarized electronic density at the muon. Our interest in the experiment considered here lies in the distribution of the magnetic field at the muon sites which have been determined to also belong to a 4*a* Wyckoff position. In the ordered phase, at 5 K, the distribution consists in a singular peak at $B^{sgl} = 91$ mT accounting for a quarter of the distribution and a continuous spectrum extending between $B^{min} = 95$ and $B^{max} = 207$ mT.

Simulations performed with the canonical helical magnetic structure [3] qualitatively confirm the features of the field distribution at the muon position [8]. A unique value for the local field is found for muons located at sites whose local symmetry axis is parallel to **k**, irrespective of the phase of the magnetic moments. For the other muon sites which are three times more frequent, the incommensurate nature of the magnetic structure leads to a continuous distribution of fields spanning between two van Hove-like singularities. The difference between the field intensity at these sites arises from the anisotropic nature of the dipole interaction. The values for the three characteristic fields $B^{sgl}$, $B^{min}$ and $B^{max}$ have the expected order of magnitude. However, the experimental value for $B^{min} - B^{sgl}$ is approximately twice the prediction for the canonical helical structure. In [9], it was shown that a dephasing $\psi = -2.04\,(11)$ degrees between the moments in the two orbits allows for a quantitative interpretation of the experimental data. Note that this phase was denoted as $\phi$ in [9].

In this paper, we discuss the robustness of the magnetic structure that was previously determined. We first illustrate that a non-vanishing dephasing angle $\psi$ is the only viable interpretation for the

experimental spectrum. Then, we show that the magnetic structure is sample independent. Finally, we demonstrate that there is no influence of the muon on the magnetic structure inferred by $\mu$SR.

## 2. Parameter Dependence of the Field Distribution

We present in Figures 1 and 2 simulations of the field distribution at the muon localization sites. For the sake of clarity, the different sources of spectral broadening which are present in the full model detailed in [9] have been set to zero. These are the spin–spin and spin–lattice relaxations, the effect of the nuclear fields and the finite correlation length of the magnetic structure. For each of the five panels, a distribution is plotted for the value of the experimental parameters used for the refinement of the zero-field spectrum recorded at 5 K [9], i.e., $k = 0.35$ nm$^{-1}$, the Mn magnetic moment $m = 0.385\,\mu_B$, $\psi = -2°$, the parameter defining the coordinates of the Wyckoff position 4$a$ for the muon $x_\mu = 0.532$, and the parameter characterizing the magnitude of the contact interaction $r_\mu H/4\pi = -1.04$. In the other two simulations shown in each panel, we successively vary one of these parameters. The main features of these simulations are the following: (i) Figure 1a indicates that the distribution is almost insensitive to the modulus of the propagation wavevector; (ii) the evolution of the distribution with $m$ (Figure 1b) is expected since the local field scales with $m$; (iii) the distribution is extremely sensitive to the muon related parameters (Figure 2). Consistently, the values found for $x_\mu$ and $r_\mu H/4\pi$ from the 5 K spectrum refinement perfectly match those deduced from transverse-field (TF) $\mu$SR measurements performed in the paramagnetic phase [8].

An overall inspection of the five panels shows that the difference $B^{\min} - B^{\mathrm{sgl}}$ significantly changes only when $\psi$ is varied; see Figure 1c. This striking sensitivity has been recently confirmed by Bonfà et al. [13]. Therefore, these simulations provide compelling evidence that the key parameter for the interpretation of the experimental spectrum is the dephasing between the moments in the two orbits. They also illustrate the remarkable complementarity of the neutron scattering and $\mu$SR techniques for the determination of the magnetic structure of MnSi. The former provides the direction and modulus of the magnetic propagation wavevector and the latter the modulus and phase of the magnetic moments.

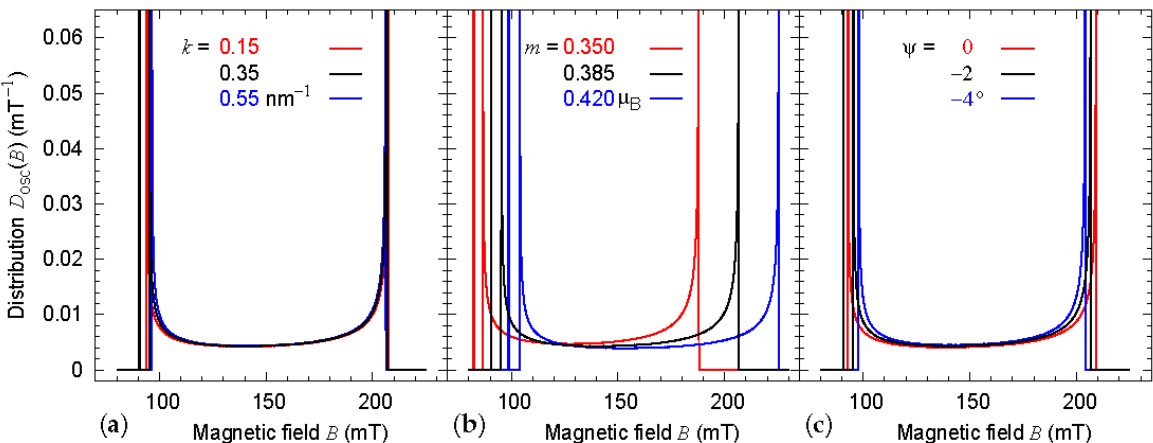

**Figure 1.** Simulations of the field distribution $D_{\mathrm{osc}}(B)$ at the muon localization sites as a function of the three magnetic parameters $k$, $m$, and $\psi$, respectively in (**a**–**c**). Note that the value of $B^{\mathrm{sgl}} = 91$ mT is essentially independent of $k$ and $\psi$.

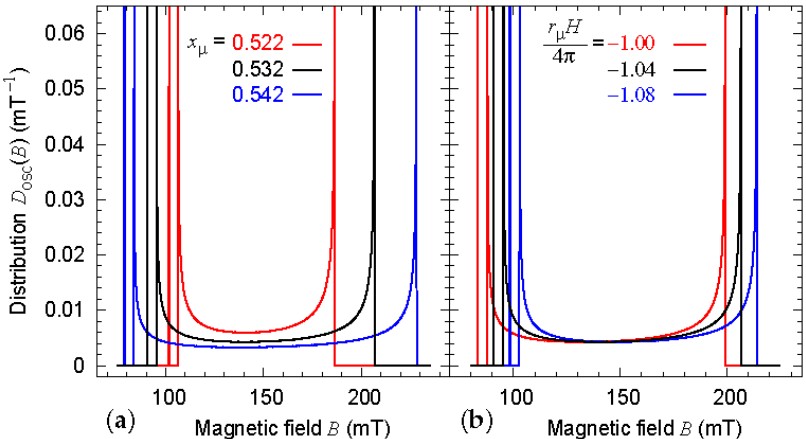

**Figure 2.** Simulations of the field distribution $D_{osc}(B)$ at the muon localization sites as a function of two muon-related parameters $x_\mu$ and $r_\mu H/4\pi$, respectively in (**a**,**b**).

## 3. Specimen Dependence of the Zero-Field Spectrum

Reliable physical measurements require samples under control. Single crystals of MnSi can be obtained using either Czochralski pulling from a stoichiometric melt or the Zn-flux method. Our published $\mu$SR measurements have all been performed with Czochralski crystals [8–10,14,15]. Neutron diffraction and thermal expansion measurements have been carried out with crystals from the same batch [16,17]. The availability of Zn-flux crystals provides an opportunity to test the robustness of the previously published magnetic structure. The metallurgical properties of the single crystals prepared by these methods are rather different as can be judged from their residual resistivity ratios: 40 versus 120 for the Czochralski and Zn-flux grown crystals of interest here. Further physical measurements performed on the latter crystals are published in [17,18]. Whereas the sizes of the Czochralski crystals are centimetric, those of the Zn-flux grown crystals are millimetric or sub-millimetric. While a piece cut from a Czochralski single crystal was enough material for the $\mu$SR measurements, a bunch of Zn-flux crystals had to be used. The latter crystals were not oriented. However, we have numerically checked that the zero-field $\mu$SR spectra are independent from the crystal orientation [9]. Unpublished measurements confirm the numerics.

In Figure 3a, we show the field distribution measured on the Zn-flux grown MnSi crystals. This distribution is obtained from a fit to the raw asymmetry spectrum resulting from the positron counts (see Section 5) using the reverse Monte Carlo algorithm supplemented by the Maximum Entropy principle. This is a model-free fit to the data, recently developed and exposed in [19,20]. In parallel, the physical model which we have evoked above and which is fully explained in [9] has been fit to the raw asymmetry spectrum. The result is shown as a full line in the graph. For reference, Figure 3b presents the same type of treatment for the data recorded on the Czochralski crystal [8,9]. Table 1 gives the parameters obtained from the physical model fit. The only notable difference for the two kinds of specimens is in the correlation length of the magnetic structure. It is found somewhat larger for the Zn-flux grown crystals than for the Czochralski single crystal. This difference may be understood by the fact that the former sample crystallizes at a temperature lower than the latter and is therefore less susceptible to structural defects. The important result of this experiment is that the dephasing between the moments in the two orbits is similar for the two types of crystals. It substantiates that this dephasing is a robust property of the MnSi magnetic structure in zero field. Pinning and defects which were supposed to affect some of the magnetic properties of MnSi Czochralski crystals [21–23] do not play a role here.

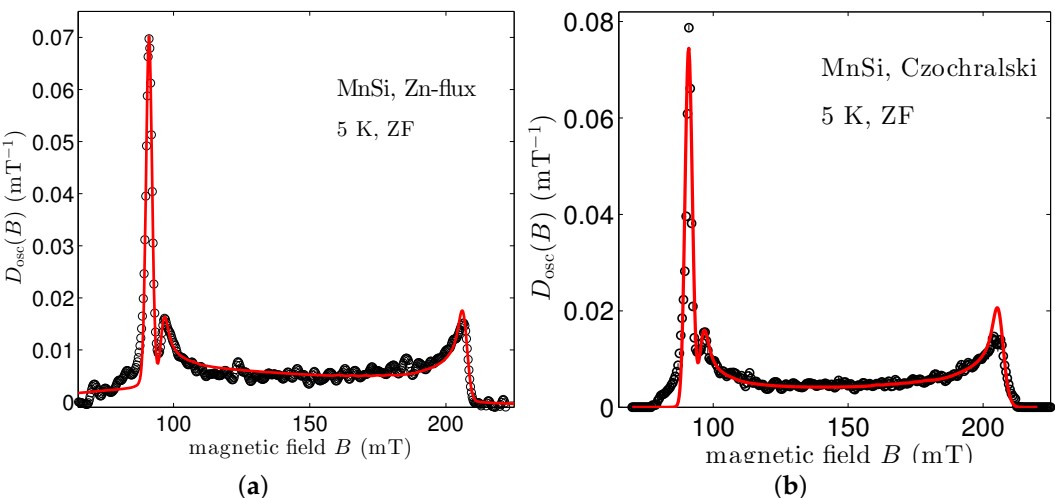

**Figure 3.** Comparison of the field distributions measured by zero-field muon spin rotation (μSR) experiments on Zn-flux grown (**a**) and Czochralski (**b**) crystals of MnSi. The circles correspond to a model-free treatment of the data according to a Maximum-Entropy supplemented reverse Monte Carlo algorithm. The uncertainties (one standard deviation) computed with this treatment are also shown: it occurs that the error bars are smaller or equivalent to the point sizes. The solid lines result from fits to the raw asymmetry spectra, i.e., they are not fits to $D_{osc}(B)$, as explained in the main text. Note that the distribution displayed in (**b**) is not obtained with the same method as that in [9], which resulted from a Fourier transform of the raw spectrum. Because of the high statistic of the spectrum and the absence of strong damping, the results are similar. It is not always the case [20].

**Table 1.** The free parameters related to the magnetic structure required to describe the μSR spectra. The rows correspond to the parameters defined in the main text and their values with uncertainties for the two samples, respectively. The parameters for the Czochralski crystal are taken from [9].

| Sample | $m$ ($\mu_B$) | $\psi$ (degrees) | $r_\mu H/4\pi$ (-) | $\xi$ (nm) |
|---|---|---|---|---|
| Czochralski | 0.385 (1) | −2.04 (11) | −1.04 (1) | 258 (35) |
| Zn-flux | 0.385 (1) | −2.11 (11) | −1.04 (1) | 391 (81) |

## 4. Testing the Possibility of a Muon-Induced Effect

The implantation of muons implies the presence of a positive electric charge in the sample under study. It may be questioned whether the presence of this charge affects its environment, especially the magnetic ions. In systems with non-Kramers spins, signatures of this effect have been observed through a modification of the crystal-field scheme of the neighbor ions. The modification is particularly effective when hyperfine enhancement [24] is at play: documented examples of a muon-induced effect in magnetic systems concern praseodymium compounds; see, e.g., [25–30]. Regarding transition element based magnetic systems, no similar effect is known. Conversely, when a detailed comparison of the magnetic phase diagram derived from μSR and nuclear magnetic resonance has been performed for the La$_{2−x}$Sr$_x$CuO$_4$ high temperature superconductors, it revealed no difference, strongly suggesting the absence of any muon-induced effect [31].

A way to probe a possible muon-induced effect in MnSi is to compare the μSR data resulting from TF experiments in a polarized paramagnetic state with the macroscopic magnetization measured in the same conditions. If the field induced at the muon sites scales with the bulk magnetization, this is a convincing indication that the properties probed by the muon are intrinsic and that no muon-induced effect is present; see, e.g., [32]. A similar comparison of local and bulk probe responses is routinely used in nuclear magnetic resonance studies to check for the influence of magnetic impurities in the bulk susceptibility or to ascertain the representative character of physical parameters obtained from the nuclear probe technique.

We first examine high TF-$\mu$SR experiments performed with $\mathbf{B}_{ext}$ applied parallel to the [111] crystal axis. For this geometry, when considering the paramagnetic phase where the Mn magnetic moments are polarized by $\mathbf{B}_{ext}$, we expect that the field probed by the muons takes two distinct values, with a population ratio 1:3 [8]. This is exactly what is observed in Figure 4.

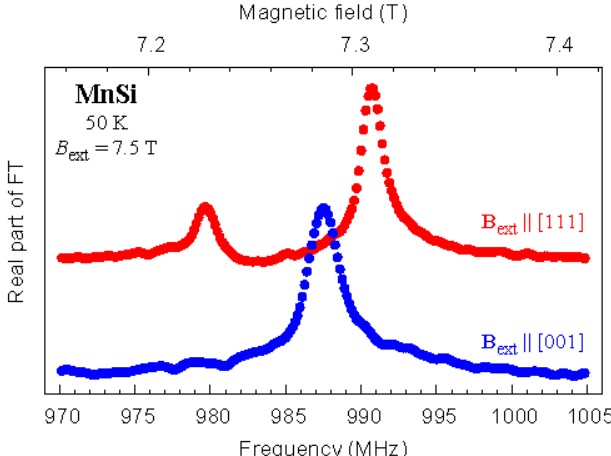

**Figure 4.** Fourier transform of transverse-field (TF)-$\mu$SR spectra recorded with the Czochralski crystals at 50 K and $B_{ext}$ = 7.5 T. The field is either set along the [111] or [001] crystal axis.

The presence of two fields is a consequence of the anisotropy of the dipole interaction and the different environments associated with the four muon sites. The splitting between the two peaks is proportional to the dipole field at the muon site and therefore to the Mn moment magnitude, and is thereafter denoted as $\Delta B_{dip}$. It is also interesting to note that the weighted average of the dipole field contribution vanishes [8]: it is therefore a simple matter to extract the contact field $B_{con}$ from data recorded with $\mathbf{B}_{ext} \parallel [111]$. Since this contact field results from the electron density at the muon site which is polarized by the Mn moments, it is also proportional to the magnitude of these moments.

When $\mathbf{B}_{ext} \parallel [001]$, TF-$\mu$SR is not sensitive to the dipole contribution from the Mn moments surrounding the muon, whatever the muon site [8]. A single value is therefore expected for the field probed by the muon, in accordance with the experiment (Figure 4). Measurements performed with this geometry give only access to $B_{con}$.

In the following, we consider the two fields $\Delta B_{dip}$ and $B_{con}$ measured for $\mathbf{B}_{ext} \parallel [111]$. In Figure 5a, we present the temperature dependence of these two fields measured at $B_{ext}$ = 520 mT. For comparison, we also plot the sample magnetization measured with a magnetometer in the same conditions. The thermal dependence of the three physical parameters matches very well in the full temperature range. Figure 5b displays $\Delta B_{dip}$ and $B_{con}$ as a function of the bulk magnetization. The experiments have been performed for different values of the temperature and field that we do not distinguish in the graph. Again, the fields measured with the muons perfectly scale with the magnetization. Altogether, the data presented in Figure 5 provide convincing evidence that the muons probe the intrinsic properties of MnSi.

There is no available experimental information on the distortion in the nearest-neighbour Mn and Si ion positions. However, a supercell density functional theory (DFT) calculation suggests the atomic displacements to be small [33].

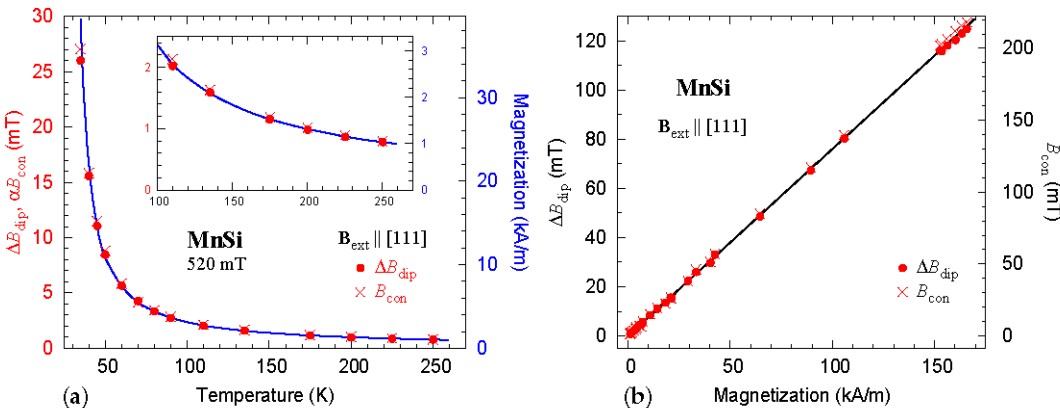

**Figure 5.** (**a**) comparison of the parameter $\Delta B_{\text{dip}}$ characterizing the muon dipole field (red bullets), the contact field ($B_{\text{con}}$, red crosses) and the sample bulk magnetization (blue line), as a function of temperature. The different parameters have been measured in a field of 520 mT applied along the [111] crystal axis. The inset displays the high temperature details. The coefficient $\alpha = 0.585$ in the vertical axis label is the ratio $\Delta B_{\text{dip}}/B_{\text{con}}$ expected from the experimental data refinements published in [8,9]; (**b**) the same parameters $\Delta B_{\text{dip}}$ and $B_{\text{con}}$ are plotted versus the bulk magnetization. The points correspond to data recorded in various temperatures and fields up to 7.5 T. The scales for the $\Delta B_{\text{dip}}$ and $B_{\text{con}}$ axes differ by the factor $\alpha = 0.585$. The full line represents the linear dependence expected if no muon-induced effect is present.

## 5. Materials and Methods

The Czochralski pulled and the Zn-flux grown crystals used in this study were prepared following procedures described elsewhere [16,17]. The $\mu$SR spectra have been recorded at the GPS and HAL-9500 spectrometers of the Swiss Muon Source (Paul Scherrer Institute, Villigen, Switzerland) with standard zero or tranverse-field setups. A zero-field measurement consists in detecting positrons in counters set parallel and antiparallel to the muon initial polarization. For a TF measurement, the positron counters are in a plane perpendicular to the external magnetic field $\mathbf{B}_{\text{ext}}$ and the muon initial polarization lies in this plane. The positrons result from the decay of polarized muons implanted into the sample under study. The anisotropy of the muon decay gives access to the muon spin evolution. This enables to probe the magnetic field at the muon position. More information can be found in [34–37].

## 6. Conclusions

We have carried out a critical analysis of previous conclusions drawn from a quantitative interpretation of $\mu$SR measurements about the magnetic structure of MnSi in zero field [9]. It has consisted in (i) examining the separate influence of the physical parameters on the field distribution measured by $\mu$SR, (ii) performing a new measurement on a sample obtained with a completely different technique, and (iii) checking the results for muon-induced effects. These three different tests confirm that the magnetic structure of MnSi in zero-field is not the conventional helical phase. They also give further credit to another study performed in an applied field, which concluded that the magnetic structure of MnSi departs from the regular conical phase [10]. A future challenge is in the comparison of $\mu$SR spectra measured in the magnetic skyrmion phase and the available models for the magnetic texture in this phase. Motivating experimental data for $Cu_2OSeO_3$ have already been published [38].

**Author Contributions:** The experiments were performed by P.D.R., A.Y., A.A., and D.A. with assistance from T.G. and R.S. for the HAL-9500 measurements. The samples were prepared by G.L. The data analysis was performed by P.D.R., A.Y., A.A., A.M. and D.A with contributions from B.R. P.D.R. and A.Y. wrote the first version of the manuscript, which was subsequently reviewed by all the authors.

**Funding:** Part of this work was performed at the Swiss muon source of the Paul Scherrer Institute, Villigen, Switzerland. D.A. acknowledges partial financial support from the Romanian UEFISCDI project PN-III-P4-ID-PCE-2016-0534.

**Conflicts of Interest:** The authors declare no conflict of interest.

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
