# Peer review of "On the Robustness of the MnSi Magnetic Structure Determined by Muon Spin Rotation"

_qubs, doi:10.3390/qubs2030019_

Round 1
Reviewer 1 Report
This is an excellent paper which summarises some very careful muSR experiments and analysis of the muon site and field distribution in MnSi. This paper should be published.
The Wycoff position of the muon is given, but given the authors are considering muon-induced effects, it would be helpful to quantify the level of distortion in the nearest-neighbour Mn and Si ions in this structure. How far do they move? What happens to the nearby bond-lengths? I suspect it will be small effects, but it would be useful to quantify this.
The paper is quite well written, but would benefit from another proofread. I spotted a few errors:
Abstract: mascroscopic should be macroscopic
end of page 4: Czochralki (typo)
page 5: praseodimium should be praseodymium
Author Response
We are grateful to the referee for the overall appreciation of the manuscript.
It is experimentally difficult to assess the muon-induced distortion. However recent DFT calculations have addressed the question. The result is that the Si displacements are less than 0.1 angstroem. The Mn displacements are predicted to be at least a factor 5 smaller.
In the resubmitted manuscript, we have added two sentences at the end of Sect. 4 to mention the existence of these recent predictions (Ref. 33) for the displacements of the atoms located in the vicinity of the muon. We have also corrected the typos spotted by the referee.
We thank the referee for the careful reading and for the suggestion of a discussion of the muon-induced displacements of the neighbor atoms.
Reviewer 2 Report
The manuscript is one in a series by this research group investigating the detailed response of the muon probe to magnetism in MnSi. In the present manuscript, the intention is (i) to test previous claims regarding the parameters in the models, (ii) test for any dependence on sample type and (iii) provide evidence against any muon-induced effects. As a result, the manuscript cannot be said to be novel, but rather addresses a well-known problem in a thorough manner. (I do not regard this as being a problem in this case!)
Each of the topics is addressed with characteristic thoroughness and compelling evidence is provided that is hard to criticise. Perhaps the final point regarding muon-induced effects is most significant here - several years ago there was much discussion of this point in MnSi. However, with the evidence provided here and elsewhere, it would be difficult now to argue that the muon significantly affects its surroundings, in contrast with the other cases the authors highlight, where the question is still an open one.
In conclusion this is a detailed study that appears, laudably, to be both thorough and correct. It will be of interest to users of the muon method in providing examples of techniques that can be used to investigate magnetic structure and to test for muon-induced distortions to the local magnetism. The paper also provides further evidence that this group's previous work on MnSi was indeed correct. I therefore recommend the paper for publication in its current form.
Author Response
We are grateful to the referee for the report and are happy that our manuscript is well received.